# Community Concern about the Health Effects of Pollutants: Risk Perception in an Italian Geothermal Area

**DOI:** 10.3390/ijerph192114145

**Published:** 2022-10-29

**Authors:** Elisa Bustaffa, Olivia Curzio, Fabrizio Bianchi, Fabrizio Minichilli, Daniela Nuvolone, Davide Petri, Giorgia Stoppa, Fabio Voller, Liliana Cori

**Affiliations:** 1Unit of Environmental Epidemiology and Disease Registries, Institute of Clinical Physiology, National Research Council, Via Moruzzi 1, 56123 Pisa, Italy; 2Unit of Epidemiology, Regional Health Agency of Tuscany, Via Pietro Dazzi 1, 50141 Florence, Italy; 3Department of Clinical and Experimental Medicine, University of Pisa, Via Roma 67, 56126 Pisa, Italy; 4Unit of Biostatistics, Epidemiology and Public Health, Department of Cardiac, Thoracic, Vascular Sciences and Public Health, University of Padova, Via Loredan 18, 35131 Padova, Italy

**Keywords:** geothermal area, risk perception, cross-sectional study, risk communication, environmental monitoring

## Abstract

Geothermal fluids for electricity and heat production have long been exploited in the Mt. Amiata area (Tuscany, Italy). Public concern about the health impact of geothermal plants has been present from the outset. Several factors influence the way people perceive risk; therefore, the objective of the present research is to develop indicators of risk perception and assess indices differences in relation to some questionnaire variables. A cross-sectional survey was conducted in the Amiata area on 2029 subjects aged 18–77. From the questionnaire section about risk perception from environmental hazards, four indicators were developed and analysed. A total of 64% of the subjects considered the environmental situation to be acceptable or excellent, 32% serious but reversible, and 4% serious and irreversible; as the values of the various perception indicators increased, an upward trend was observed in the averages. Risk perception was higher among women and young people, and was associated with higher education. Those who smelled bad odours in their surroundings reported higher risk perception. Furthermore, risk perception was higher in four municipalities. The results represent the basis for further investigations to analyse the link among risk perception indicators, exposure parameters, and health status.

## 1. Introduction

In recent years, interest in the exploitation of geothermal fluids has grown worldwide. Considered an important renewable energy, a doubling of geothermal electricity generation and a fivefold increase of geothermal heat has been planned for Europe in the next ten years [1]. Eighteen EU countries have included geothermal energy in their 2020 National Renewable Energy Action Plans: twelve countries consider both geothermal heat and electricity, while six consider only heat [2].

In Italy, all geothermal power plants in operation are located in Tuscany Region, the region that most of all represents Italian geothermal energy. The exploitation of this kind of energy began in the first half of the nineteenth century but in reality, historical sources have revealed that the heat of the earth and natural springs have been exploited since ancient times, at least for the entire first millennium BC by the Etruscan populations. Furthermore, these geothermal resources have proved to be among the most productive in the world [3,4]. Starting with Larderello, which today houses the largest geothermal plant in Europe, over the decades the number of regional geothermal plants has grown to more than thirty. Most of the Tuscan geothermal energy comes from the heat that derives from the intrusion of a magmatic pluto under the volcanic complex of Mt. Amiata. There are many other areas with abundant geothermal resources in Italy, in regions such as Veneto, Friuli-Venezia Giulia, Campania, Sicily and Emilia Romagna; however, areas external to Tuscany have almost no relevance as an impact in absolute terms on the Italian energy balance. Currently, 36 geothermal plants are distributed in the Provinces of Pisa, Siena and Grosseto and, in 2019, within an overall regional production of 16,566 Gigawatt-Hour (GWh), the contribution of the geothermal power plants amounted to 5688 GWh, thus covering 34% of the regional electricity requirement [5] (Figure 1).

Particularly, monitoring activities are conducted by the Geothermal Sector of the Regional Agency for Environmental Protection of Tuscany (ARPAT). All monitoring data are reported through annual reports (http://www.arpat.toscana.it/documentazione/report/report-geotermia/report-sul-monitoraggio-nelle-aree-geotermiche, accessed on 27 July 2022). Specifically, in 2019 no overruns of the emission limit values for the authorized parameters (mercury, hydrogen sulphide (H_2_S) and sulfur dioxide) were detected. The efficiency of ammonia and H_2_S abatement systems was higher than the minimum limit value [6].

The Regional Health Agency of Tuscany (Agenzia Regionale per la Sanità-ARS) is in charge of providing scientific support to policy makers by producing epidemiological studies on population health status and healthcare services. For several years ARS has been monitoring the health of the populations living in the Tuscan geothermal areas. The results of descriptive studies conducted by ARS in collaboration with the Institute of Clinical Physiology of the National Research Council of Pisa (IFC-CNR) [7,8], have highlighted some mortality and hospitalization excesses in the Amiata area, in particular for liver and stomach cancers and respiratory diseases, using as a reference the rates observed in an area of 98 municipalities falling within a circle of 50 km radius pointing to the centroid of the area under study. The etiological study by Nuvolone et al. (2019), while confirming the excesses of mortality and hospitalization for respiratory diseases, also revealed a mortality risk decrease from ischemic heart diseases and cerebrovascular diseases in relation to high exposures (20–33 µg/m^3^) to H_2_S emissions [9].

It should be pointed out that Mt. Amiata area, as well as being characterized by a complex natural hydrogeological context, has been the focus of various anthropic activities over the decades. In fact, for about a century, there was an intense extraction of cinnabar, the sulphide from which mercury was obtained, and mercury industrial production, which is known for its negative impact on human health [8,10].

In recent years, several studies have been conducted on the health status of the Amiata communities [8,9,11]; during this period, public concerns about the health consequences of long-term geothermal energy exploitation were expressed through public statements, social media and traditional media dissemination, and participation in public conferences [1,12]. In addition, attention has grown in consideration of national and European policies on renewable energy, which propose a growth in geothermal energy exploitation in Europe, as an alternative to the traditional fossil-fuelled plants [2,13]. In this context, the centrality of citizen participation was highlighted [14], and both the public authorities in charge of health protection and the researchers involved in the Amiata area maintained a continuous relationship with local communities, citizens’ associations, and public administrators, in order to inform them about the activities and to involve them in the discussion of the research results.

The InVETTA project (Biomonitoring Survey and Epidemiological Evaluations for the Protection of Health in the Amiata Territories), performed by ARS, carried out a human biomonitoring study by collecting blood and urine samples from 2060 subjects and a questionnaire on habits, living and working environment, and clinical history, including a section on risk perception [15,16]. This latter specific section of the questionnaire has been used in recent years to complement the human biomonitoring research with findings on the people involved, their attitudes towards risks and hazards, and sources of information in different parts of society [17,18,19,20,21]. Particularly, here we are going to report and discuss results based on the InVETTA section questionnaire on risk perception. This work is based on the assumption that an analysis of risk perception and access to environmental information by various institutional and non-institutional actors in Amiata communities deserves specific interest in support of public policy, health promotion, environmental protection and citizen participation in territorial management. In fact, risk perception is relevant and related to risk communication, encompassing the level of awareness in a specific area, comprehension and history of the community itself, including the ability and willingness to deal with risk [22]. The direct participation of people and communities in polluted areas is increasingly appreciated and considered relevant to promote policies for anthropogenic risk mitigation and environmental sustainability. Experience has shown that improvements can be too slow and risk governance ineffective without direct citizen involvement, including a control and monitoring function [23]. Anthropogenic hazards seem to be more acceptable than natural hazards and are very related to social acceptance, knowledge, and awareness of risk [24]. Social acceptance is crucial to address shared solutions based on trust in public authorities [25]. Therefore, it is valuable to share the results of scientific investigations, understand population perceived risks and engage communities to achieve sustainable environmental and health protection measures [17,19,26,27].

Risk perception, in general terms, is a cognitive process that guides people’s behaviour when faced with decisions involving potential risks. Risk perception involves two main dimensions: a cognitive dimension, linked to knowledge and understanding of risk, and an emotional dimension, which includes feelings; both are components of the reaction to risks, representations of immediate and/or future consequences and their implications, and the way people determine how to behave accordingly [28]. The psychometric paradigm has helped to elucidate how certain elements and characteristics are specifically influential in the perception of the hazardousness of an activity, such as direct control over hazardous events, voluntariness, the size, and scope of consequences [29]. Several models and “heuristics” have been proposed to examine collective and individual responses to risks, which are useful to interpret to propose and promote effective risk reduction strategies [26,30,31,32].

The objective of the present paper was to develop risk perception indicators specifically related to the geothermal context, to study their characteristics, and to verify the association of these indicators with variables included in the questionnaire referring to socio-demographic characteristics, evaluation of the environmental situation, self-reported personal exposure to pollutants and place of residence.

## 2. Materials and Methods

### 2.1. The Survey

The study sample of the InVETTA project consisted in subjects aged 18–77 residing in the municipalities in the Amiata area most affected by geothermal plant emissions (Abbadia San Salvatore, Piancastagnaio, Arcidosso, Santa Fiora, Castel del Piano and Castell’Azzara defined as “Main area”) and in a control group of municipalities, always belonging to the geothermal area (Seggiano, Radicofani, Cinigiano and Castiglione d’Orcia), defined as “Control area”. The distinction between “Main area” and “Control area” is based on the atmospheric H_2_S concentration values estimated with dispersion models created using internationally standardized software and validated by the monitoring stations present in the area [9]. All further information on materials and methods of the InVETTA project can be found in the dedicated report published in 2021 [16]. Although the subjects have mainly been enrolled through the municipal registry office lists, a sub-sample is composed of both volunteers and workers in ENEL, the energy company owner of geothermal plants. Furthermore, sampling was stratified by age, sex and H_2_S exposure. Participants in the study were asked to fill in a questionnaire on habits, living and working environment, personal clinical history, and risk perception [16].

### 2.2. The Questionnaire Section on Risk Perception

The questionnaire used during the InVETTA project was designed based on previous human biomonitoring surveys, tested to assess the sustainability of the total number of questions, individual and overall comprehensibility [15,17,18,19,20]. The questionnaire consisted of 12 sections (105 total questions), collecting data regarding socio-economic status, exposures, behaviours and lifestyle, clinical history, and risk perception. The last section included questions exploring risk perception and sources of information. The questions were structured in a closed format and accompanied by a series of options. A value judgment was required and a Likert scale with five alternatives was used. The questions relating to the perception of territorial characteristics that generate concern were formulated as hazards, also providing a definition: a series of hazards were listed, in relation to which a value judgment and application to one’s own person and area of residence was requested (You are faced with a list of different hazards. To what extent do you personally feel exposed to each of them? Extremely; Very; Moderately; A little; Not at all). With regard to risks, the questions dealt with sources of information, responsibilities for health protection and the environmental situation in the area of residence (Excellent, Acceptable, Serious but reversible, Serious and irreversible). Two further sets of questions asked for a value judgment on the possibility of falling ill from a range of diseases while living in the area of residence and the same diseases in a generic polluted area: these were the questions that directly provided information on each person’s risk perception.

The definitions of hazard and risk were specified in a note in the questionnaire section number 12.

Definition of hazard-Hazard is a potential source of harm. It is defined on the basis of intrinsic properties or characteristics of an object or situation, which may cause undesirable consequences. Hazard is not measured, but it is a property. Examples of hazards: an industrial plant; inorganic arsenic in water.

Definition of risk-Risk is the probability that personal injury (health, environmental, economic) will result from exposure to a hazard. It is a quantitative measure. Example: the mortality risk for exposure to ultrafine particles increases by 7% for each 10 µg/m^2^ increase in ultrafine particles.

This section aimed to detect risk perception, vulnerability of the territory, community awareness of existing problems, sources of information and actors considered responsible for health protection. The questionnaire administration and data collection were carried out between January 2017 and May 2019.

Figure 2 shows the methodology of the InVETTA project and our focus on risk perception.

### 2.3. Development of the Risk Perception Indices (RPIs)

In order to construct the RPIs, the study by Signorino and Beck (2014) was taken as a reference for outlining the perception profile of the population [33]. The research object of this work was based on the answers to the questionnaire, which were considered adequate tools to detect the dimension, also subjective, of the object of the study. The main parameters and the appropriate indicators that allow the measurement of the perception of risk and hazards in this area were identified.

The indexes used were the following:

the Hazard Perception Index (HPI);

the Exposure Hazard Perception Index (EHPI);

the Health Risk Perception Index (HRPI);

the Risk Perception Index (RPI).

Each index was calculated using the formula proposed in Signorino and Beck, (2014), for a sample of N respondents to which, for each environmental/health risk, it is required to express the degree of concern on a Likert scale, according to the following formula:$$\mathrm{HPI},\;\mathrm{EHPI},\;\mathrm{HRPI},\;\mathrm{RPI}=\frac{\sum_{i}^{k}n_{i}\pi_{i}}{N\cdot \left(k\right)}$$
where:

n_i_ was the absolute frequency of responses in the i-th response mode;

π_i_ was the weight assigned to the i-th mode (example: 0—Not at all; 1—A little; 2—Moderately; 3—Very; 4—Extremely);

N was the total number of observations (coinciding with the number of respondents);

k was the weight of the major class of the Likert scale (in the example equal to 4) [33].

Each indicator can assume values between 0 and 1: the closer the value is to 1, the higher the risk perception is. HPI, EHPI, HRPI, and RPI were calculated for each participant. Spearman’s rank correlation test was used to check for pairwise correlations between each pair of indices.

To calculate these indices, questions concerning the state of the environment and the perception of environmental and health risks were selected from the questionnaire (Table 1).

Informative questions were selected on the perception of environmental hazards and health risks related to geothermal energy, in particular: the personal connotation of exposure to the different hazards, the exploration of the perception of territorial aspects in personal exposure in relation to the area of residence, and the assessment of the environmental and health situation in the municipality of residence.

### 2.4. Statistical Analyses

The questionnaires were first digitized through a web mask produced by ARS and then analysed through descriptive analyses.

Perception indicators were described through the mean, the Standard Deviation (SD), the minimum and maximum, the 25th (25p), 50th (50p), 75th (75p) and 90th (90p) percentiles. Since the distributions of the perception variables were not normal, the comparison of the averages of the different perception indices among the study factors categories (i.e., gender, age classes, educational qualification in classes, being/not being voluntary, presence/absence of odours, municipality of residence and exposure to dust, chemicals, pesticides, gases or radiation) and question 12.5 (How do you judge the environmental situation of the Municipality where you live in?), was carried out by means of the non-parametric Kruskal–Wallis equality-of-populations rank test considering one factor at a time, accompanied by the *p*-value. The correlations between the perception indices HPI, EHPI, HRPI and RPI were calculated using Pearson’s correlation index accompanied by the *p*-value. For all indices, we considered an ordinal scale of perception divided into 5 classes: 0.2 ≤ low, 0.2–0.4 low-medium-, 0.4–0.6 medium, 0.6–0.8 medium-high and high if ≥0.8.

The outliers were defined using the classical statistical method and were all those that exceeded the value of the following formula:Q3+1.5(Q3−Q1)
where:

Q3 is the 75p;

Q1 is the 25p.

The limit of statistical significance was set at *p* < 0.5.

### 2.5. Ethical Aspects

The study was approved by the Regional Ethics Committee for Clinical Trials of the Tuscany Region (Registration number 10679_2018/), responsible for the study area. Participation in the study was voluntary and no incentives were offered. All participants received written and oral information about the study and signed the informed consent form for research purposes. All data were collected and analyzed in accordance with the Italian Law n. 196 of 30/6/2003 (“protection of personal data”) and subsequent amendments, in fully compliance with European directives about citizens’ privacy.

## 3. Results

The questionnaire administration and data collection were carried out between January 2017 and May 2019. Questionnaires were firstly digitized through a web mask created by ARS and then analysed. Table 2 shows the characteristics of the 2029 respondents to the questionnaire.

Almost half of the subjects (47%) were resident in the two municipalities of Piancastagnaio and Abbadia San Salvatore, and 62% in the municipality of Arcidosso, belonging to the Main area.

Of the 2136 subjects selected, 2029 (93%) responded the questionnaire: 1027 (50.6%) subjects were extracted from the municipal registries, 978 were volunteers (48.2%) and 24 (1.2%) were ENEL workers.

For the 24 ENEL workers, a pre-analysis of risk perception was carried out to check whether they were similar to or different from the residents. Since the risk perception indices were significantly lower than the average of the rest of the participants shown in Table 3 (HPI 24 workers: mean = 0.28; SD = 0.20; EHPI 24 workers: mean = 0.32; SD = 0.12; HRPI 24 workers: mean = 0.31; SD = 0.16; RPI 24 workers: mean = 0.28; SD = 0.13), the 24 workers were excluded from the subsequent analyses to avoid selection bias.

Of the remaining 2005 questionnaires, 137 (6.8% of the total) contained missing values in at least one of the variables used to construct the perception indices, 199 in the case of HRPI. Therefore, descriptive analyses were performed on 1868 questionnaires for HPI, EHPI, RPI and on 1806 for the HRPI. Table 3 shows the descriptive statistics of the four perception indices. The mean of each of the indices was statistically different from the other (*p* < 0.05), in particular, the mean of HRPI was higher than of both HPI and EHPI, which were more similar to each other (Table 3).

We observed that the mean values of hazard and exposure perception were lower than the values of health risk and total risk perception (Table 3). Only four respondents resulted as outliers.

Indices were all significantly correlated with each other, but the strength of the correlation changed according to the indices considered: HPI strongly correlated with EHPI and RPI, moderately with HRPI, while RPI was strongly correlated with the other three indices (Table 4).

Table 5 shows the descriptive statistics of HPI, EHPI, HRPI and RPI by sex, age groups in quartiles, education level in three classes (education less than or equal to middle school, education equal to high school, education equal or higher than university), being/not being a volunteer, presence/absence of odour perception, main area/control area, municipality of residence and occupational exposure to dust, chemicals, pesticides, gas or radiation.

As far as perception indices are concerned, the following considerations can be made (Table 5):-women had a significantly higher perception than men (HPI 0.08, EHPI 0.06, HRPI 0.06, RPI 0.07);-with increasing age, the HPI, EHPI, RPI decreased significantly;-up to the age of 59, EHPI, HRPI and RPI remained constant and then decreased significantly in older subjects;-as education level increased, all types of perception increased significantly;-volunteers had a significantly higher perception of risk (HPI 0.05, EHPI 0.03, HRPI 0.04, RPI 0.04);-subjects who reported perceiving unpleasant odours had a significantly higher perception of hazard/exposure/risk (HPI 0.08, EHPI 0.08, HRPI 0.05, RPI 0.06);-subjects living in the Amiata municipalities most exposed to geothermal emissions (main area) had a significantly higher perception of hazard/exposure/risk (HPI 0.06, EHPI 0.06, HRPI 0.06, RPI 0.07);-subjects reporting occupational exposure to dust, chemicals, etc., had a significantly lower perception of hazard (HPI 0.04) exposure (EHPI 0.02) and risk (RPI 0.02).

The description of the four indicators by municipality showed an overall higher perception of hazard and risk in the municipalities of Abbadia San Salvatore, Arcidosso, Castel del Piano, Piancastagnaio and Seggiano, all belonging to the Main area except Seggiano (Table 6).

Approximately 64% of the sample considered the environmental situation to be acceptable or excellent, while the remaining 36% perceived it to be severe, but of these, only 4% considered the situation as irreversible (Table 7). For all perception indices, both hazard and risk (all between 0.4 and 0.6), the average perception showed a significant increasing trend from excellent to severe-irreversible (Table 7).

Table 8 shows the frequency distributions of the answers to question 12.5 according to sex, age group, being a voluntary participant or not, belonging to the main or control area, and residence in one of the six most exposed municipalities of the main area. The self-reported assessment of the environmental situation was quite different between the two genders, in particular, a lower percentage of women than men considered the situation to be “excellent” (3.9% vs. 10.6%), on the contrary, the situation was evaluated as severe, reversible or not, more among women (39.9% vs. 31.7%) (Table 8).

Considering the age groups, the differences were not statistically significant although the 18–39 years old tended to appear more concerned than the elderly (Table 8). Volunteers rated the environmental situation as more severe, reversible, or irreversible, than non-volunteers (44.8% vs. 29.1%) (Table 8). Responses on the environmental situation were differentiated according to residence in the main or in the control area, with the former rating the situation as more serious (41.1% vs. 21.1%) (Table 8). Analysing the answers in the six municipalities of residence, a more serious environmental situation is declared by the sample residing in the municipalities of Abbadia San Salvatore and Arcidosso (Table 8), main area.

## 4. Discussion

The objective of the present paper was to develop environmental and health risk perception indicators specifically related to the geothermal context, study their characteristics, and verify their association with variables included in the questionnaire referring to socio-demographic characteristics, evaluation of the environmental situation, self-reported personal exposure to pollutants and place of residence.

The main parameters and indicators allowing the measurement of hazard and risk perception in the Amiata area were identified and processed, creating four risk perception indicators (HPI, EHPI, HRPI and RPI). In order to calculate these indicators, questions concerning the state of the environment and the perception of environmental and health risks were selected, particularly those relating to the hazard and risk perception associated with geothermal energy production. Since the research was based on the responses to questionnaires from the Amiata population, it was possible to detect the subjective dimension of the risk perception features. The salient issues investigated and identified were the awareness of exposure to different hazards, the perception of surrounding environment and personal exposure, as well as the health risk perception.

About 64% of the subjects involved in the research considered the environmental situation acceptable or excellent. Only about 4% believed that the situation was serious and irreversible, showing, however, a moderate perception of hazard, environmental exposure and health risk. The analyses conducted in the present paper revealed that the perception of risk and hazard was higher among women and young people and was associated with a higher education level. Volunteers had a higher perception, as well as those who noticed bad smells around their homes. The sampled subjects living in the municipalities of Abbadia San Salvatore, Arcidosso, Piancastagnaio and Seggiano, all belonging to the main area except the last one, showed a higher perception.

As far as we know, this is the first study that analyses the perception of environmental and health risk in relation to the geothermal phenomenon, calculating risk perception indicators and comparing them. Given the current lack of previous studies in geothermal areas conducted using perception indices, a comparison was made with the same indicators calculated in two high-risk areas with active petrochemical plants (Milazzo and Priolo, in Sicily Region), one high-risk area with abandoned industries (Crotone, in Calabria Region), and three non-polluted reference areas [21]. The HPI in Amiata showed a lower value than in the two areas with active industries, and a higher value than in Crotone. The EHPI, HRPI and RPI in Amiata showed a rather low average value compared to the other high-risk areas, especially compared to the two active industrial areas. It should be noted that the perception indices calculated in the main Amiata area were more similar to those of the site without emission sources (Crotone) compared to the two sites with emitting petrochemical plants, and always lower than the value in the reference area of the three high-risk areas. The Italian SEpiAs project [7] compared community exposure to arsenic, through human biomonitoring and a questionnaire, in two areas affected by predominantly natural contamination, and two of industrial origin. One of the studied areas was the Amiata area, where the presence of natural arsenic was well known and monitored in wells and drinking water. Risk perception was investigated using the same section of the questionnaire here analysed. The results showed a greater awareness of contamination in industrial areas, while the Amiata respondents in particular showed a risk perception similar to that of industrial areas, concerned in particular about the risk of cancer, respiratory diseases and leukaemia from environmental pollution [19]. In other studies, using the same questionnaire, the most significant questions concerned personal exposure to pollution and the risk of health consequences of this pollution. The most feared outcomes were cancer and congenital anomalies, as related to the impact on the surrounding environment, where the situation was defined in most cases as “serious and reversible” but often also “serious and irreversible”. The samples investigated showed different levels of risk perception depending on the specific situation and history of the territory, exposure to information, direct experience, and social influences [17,18,20]. Generally, anthropogenic risks seem to be more acceptable than natural risks even if they are related to public and social acceptance that in turn are linked to risk management [24]. Our survey also revealed a higher risk perception among both women and young subjects. These findings were consistent with other research suggesting men were more likely to agree with the geothermal technology than women [34] which usually were more concerned with the technology risk [35] and focused more on advantages and disadvantages that could concern the community [36]. This higher risk perception of women and younger, also associated with higher education, can be found in other research about risk perception, as we found a connection with knowledge, awareness, and social acceptance [14,24]. According to the four identified indices, the awareness of environmental disturbance, such as unpleasant odours, was linked to a higher risk perception. On the other side we found that subjects who reported occupational exposure to dust, chemicals, etc., had a significantly lower perception of both hazard and exposure, which is consistent with the observation that a subject who has constant experience of a risk is more inclined to accept it, especially if it is part of his/her job and if other social and economic considerations come into play [24,28].

The social role of participants, their ability, and possibilities to act to modify the personal and community situation contribute to defining risk perception [24,28]. The human biomonitoring donors and questionnaire respondents which voluntarily participated in the InVETTA study, about half of the total respondents, had a significantly higher risk perception according to the four indicators analysed: we could argue that one of the reasons of their participation in the study was their concern for the environmental situation. The answers in the Amiata municipalities were not uniform in geographical terms: a more worrying environmental situation was declared by the residents in Abbadia San Salvatore and Arcidosso. In future analyses, more in-depth evidence should be gathered to understand these differences, which could be attributed to the more intense exposure to information and protests against existing plants in the past, and the experience of disturbances, such as odours or water pollution attributed to geothermal plants [1,12,37].

As regards the limitations of the InVETTA study, the main one is represented by the relatively modest 41.6% on average participation of sampled subjects. This could be possibly due to the recent proliferation of requests for participation in epidemiological surveys, also coupled with the increasing scientific complexity and a background of growing distrust in science and institutions [38,39]. Low participation may have introduced a bias resulting from respondents not being as representative of the population as invitees. Indeed, there is a risk that individuals with specific characteristics, such as those who for example those who have a greater risk perception, responded. 48% of the total number of participants were volunteer subjects, citizens who applied spontaneously. While the subjects sampled tended to represent a random group of residents, volunteers may be selected because they are at higher/minor risk for both exposure and risk perception. Furthermore, in the InVETTA study all indices of risk perception were lower for sampled subjects than for volunteers, confirming for the latter a greater concern and sensitivity with respect to environmental issues. This different sensitivity on environmental quality of volunteers is presumably related to the fact that their percentage is significantly higher in the main municipalities than in the control municipalities. In addition, in the present study we did not consider the length of time the respondents have lived in the given area and whether or not they own the property, two factors known to have some impact on the perception of environmental risk. Also, whether respondents had/had not children, another known characteristic that affects risk perception because of concerns about the impact on children’s health, was also not included among the variables of interest. This study has an exploratory nature, and it will be important for future research to analyse these issues in more detail.

The study also presents some strengths, such as the fact that the sample of subjects invited into InVETTA was representative of the 18–70-year-old population residing in the area. In addition, as is the standard practice in this type of survey, there was a substitution procedure to make up, in part, for the low participation. Moreover, comparison between those who agreed to participate and those who refused showed no significant differences and the participation of volunteer subjects was widely encouraged in order to enhance citizens’ interest in this public health initiative. The risk of introducing bias through the inclusion of volunteers was minimized at the analysis stage by considering information on participation mode and conducting stratified analyses in the two subgroups of sampled subjects and volunteers.

## 5. Conclusions

The results obtained from this analysis provide a picture of the perception of environmental and health risks, particularly those arising from the exploitation of geothermal energy, that are of greatest concern to the communities of Mt. Amiata, where production plants have been present since the 1960s. The results of this study are important since they allow for a better understanding of the situation in the geothermal area, particularly on the concern and risk perception of the resident communities. In fact, the results of human biomonitoring surveys alone can create further anxiety; awareness of the different sensitivities present in the area is a means of building trust over time, identifying appropriate public interventions and accompanying prevention measures. Social acceptance is crucial for understanding and addressing shared plans and solutions, and the single most influential factor seems the trust in public authorities and politics [25]. Indeed, it is necessary to share the results of scientific surveys, to understand the anxieties and perceived risks, and to involve communities in order to achieve sustainable measures that guarantee the protection of health and the environment [17,19,26,27], and this study reinforces this concept. Informed and mature risk governance on the part of the relevant authorities includes a willingness to engage in constructive and inclusive dialogue, capable of openly negotiating the needs of different stakeholders, putting public health and the environment at the centre [40] in order to build risk communication programs. Risk communication, in fact, is an activity that must be carried out on the basis of an accurate knowledge of local risk perception. This is essential to identify major concerns and avoid creating more anxiety in the community, to enable shared decision-making on preventing existing risk factors and reducing hazardous exposures. For the future, it would be interesting to explore more specifically qualitative aspects through activities based on local focus groups; this next phase of the research would be very useful to even better understand why people have a certain and specific perception of environmental risk related to geothermal energy and to the living context that people are experiencing. The present research is very focused on individual perceptions, but people share concerns/information/responses at the societal/community level, and we know that risk perception can be amplified in the social context. With this article we also want to lay the foundation for further investigations in which additional environmental, pollutant exposure and health data will be collected. We believe that studies on risk perception in areas with pollutant sources are important to follow up on the spatio-temporal evolution of environmental and health perceptions in relation to trends in measured environmental and health quality parameters.

## Figures and Tables

**Figure 1 ijerph-19-14145-f001:**
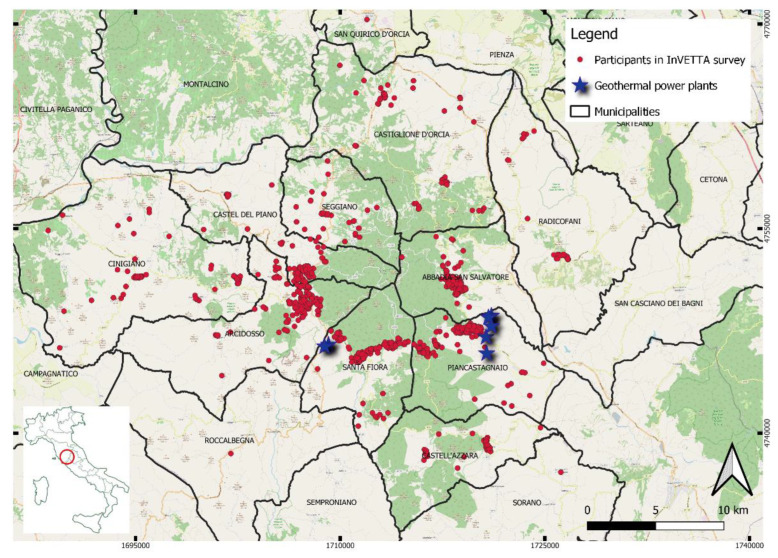
Localization on the territory of the 36 geothermal plants in Tuscany (blue stars) (Reference System Monte Mario/Italy Zone 1 (west time)–Datum: Roma40–Projection: Gauss-Boaga–west time–EPSG 3003). Note: For privacy reasons and to protect the confidentiality we applied geomasking techniques with a random shifting of the coordinates of the points.

**Figure 2 ijerph-19-14145-f002:**
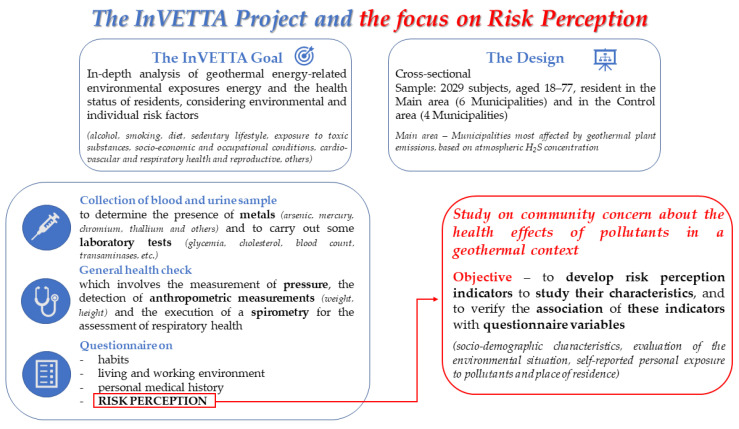
Scheme of the InVETTA project and how our study fits within the InVETTA project.

**Table 1 ijerph-19-14145-t001:** Questions used to calculate the Hazard Perception Index, the Exposure Hazard Perception Index, the Health Risk Perception Index and the Risk Perception Index.

Question Number	Question *	Answer Options for Each Sub Question	Index Defined
12.1	You are faced with a list of different hazards. To what extent do you personally feel exposed to each of them? b. Noise c. Bad smells g. Air pollution j. Water pollution k. Hazardous industries l. Earthquake m. Food contamination	0. Not at all 1. A little 2. Moderately 3. Very 4. Extremely	Exposure Hazard Perception Index-EHPI
12.2 **	Among the hazards listed above, which do you think are present in the area you live? b. Noise c. Bad smells g. Air pollution j. Water pollution k. Hazardous industries l. Earthquake m. Food contamination	0. Present 1. Not present	Hazard Perception Index-HPI
12.5 ***	How do you judge the environmental situation in your residence municipality?	1. Excellent 2. Acceptable 3. Serious but reversible 4. Serious and irreversible	
12.9	In your opinion, how likely it is, in your residence area to fill ill due to a. Allergies; b. Acute respiratory diseases; c. Chronic respiratory diseases; d. Cardiovascular diseases; e. Infertility; f. Various form of cancer; g. Leukemia; h. Congenital Malformations	0. Not at all 1. Unlikely 2. Medium probability 3. Very likely 4. Sure	Health Risk Perception Index-HRPI
12.1 12.2 12.9	These questions were used to calculate the Risk Perception Index (RPI), an overall indicator of environmental and health risk perception.		

Notes: * For each question, only the options that had a direct relationship with the geothermal phenomenon were considered for the analyses; ** Dichotomous variables transformed replacing the answer “Not present” and “Present” with the values 0 and 2, respectively, of the Likert scale used for the other questions. In this way each variable weight is uniform; *** This question was used as control variable to assess the consistency of the responses to questions 12.1 and 12.2.

**Table 2 ijerph-19-14145-t002:** Characteristics of the sample of respondents to the questionnaire.

General Characteristics	Classification	N	%
Total subjects		2029	100.0
Subjects’ origin	Municipal registries	1027	50.6
	Volunteers	978	48.2
	ENEL workers	24	1.2
Sex	Men	885	43.6
	Women	1144	56.4
Age in quartiles (years) Average = 49.3; SD = 13.7 years)	18–39	508	25.0
	40–50	507	25.0
	51–59	507	25.0
	60–77	507	25.0
Residence municipality	**Main Area**		
	Abbadia San Salvatore	465	22.9
	Arcidosso	299	14.7
	Castel del Piano	172	8.5
	Castell’Azzara	104	5.1
	Piancastagnaio	492	24.2
	Santa Fiora	243	12.0
	**Control Area**		
	Castiglione d’Orcia	73	3.6
	Cinigiano	84	4.1
	Radicofani	44	2.2
	Seggiano	41	2.0
	Others	12	0.7
Marital status	Unmarried/Maiden	645	31.8
	Married	1204	59.3
	Divorced	70	3.4
	Separated	51	2.5
	Widowed	56	2.8
	Other	3	0.1
Educational qualification	Primary school license	75	3.7
	Middle school license/Professional start-up	549	27.1
	High school diploma	1000	49.3
	Degree/University diploma	381	18.8
	Specialization/Master	19	0.9
	Other	5	0.2
Occupation	Employee	83	4.1
	Housewife	82	4.0
	Autonomous	10	0.5
	Chief worker	162	8.0
	Farmer/Breeder	24	1.2
	Senior Executive/Supervisor/High school teacher	72	3.5
	Manager	38	1.9
	Executive Employee	306	15.1
	Entrepreneur	95	4.7
	Agricultural worker	12	0.6
	Self-employed worker	213	10.5
	Freelance	125	6.2
	Generic worker	355	17.5
	Student	74	3.6
	Technician/Employee	284	14.0
	Other	94	4.6

**Table 3 ijerph-19-14145-t003:** Descriptive statistics of Hazard Perception Index, Exposure Hazard Perception Index, Health Risk Perception Index and Risk Perception Index.

Indices	N	Mean	SD	Minimum	25p	50p	75p	90p	Maximum
HPI	1868	0.40	0.28	0.00	0.14	0.43	0.57	0.71	1.00
EHPI	1868	0.39	0.20	0.00	0.25	0.39	0.54	0.68	0.93
HRPI *	1806	0.45	0.17	0.00	0.34	0.47	0.56	0.66	0.97
RPI	1806	0.41	0.17	0.00	0.27	0.42	0.53	0.64	0.91

Note—* Of the 1868 questionnaires, 62 did not answer all 12.9 a–h questions. Legend-HPI: Hazard Perception Index; EHPI: Exposure Hazard Perception Index; HRPI: Health Risk Perception Index; RPI: Risk Perception Index; N: number of questionnaires; SD: Standard Deviation; 25p: 25th percentile; 50p: 50th percentile; 75p: 75th percentile; 90p: 90th percentile.

**Table 4 ijerph-19-14145-t004:** Statistical correlations among the perception indices.

	HPI	EHPI	HRPI
EHPI (rho)	0.66		
(*p*-value)	<0.001		
HRPI (rho)	0.31	0.36	
(*p*-value)	<0.001	<0.001	
RPI (rho)	0.88	0.84	0.64
(*p*-value)	<0.001	<0.001	<0.001

Legend–rho: Pearson’s correlation coefficient; *p*-value: observed probability of accepting the hypothesis of non-correlation between perception indices or vice-versa the probability of rejecting the hypothesis of correlation; HPI: Hazard Perception Index; EHPI: Exposure Hazard Perception Index; HRPI: Health Risk Perception Index; RPI: Risk Perception Index.

**Table 5 ijerph-19-14145-t005:** Descriptive statistics of the perception indices by gender, age classes, educational level in 3 classes, volunteer or not, odours presence/absence, main area/control area, residence municipalities and occupational exposure to dust, chemicals, pesticides, gas or radiation.

Factors	N	HPI			EHPI			HRPI			RPI		
		Mean	SD	*p*	Mean	SD	*p*	Mean	SD	*p*	Mean	SD	*p*
Sex				<0.001			<0.001			<0.001			<0.001
Men	778	0.36	0.27		0.36	0.19		0.42	0.16		0.37	0.17	
Women	1090	0.44	0.28		0.42	0.20		0.48	0.17		0.44	0.17	
**Age classes (years)**				0.007			<0.001			0.041			<0.001
18–39	484	0.43	0.28		0.41	0.18		0.45	0.18		0.43	0.17	
40–49	481	0.42	0.27		0.41	0.20		0.46	0.17		0.43	0.17	
50–59	469	0.40	0.27		0.40	0.20		0.46	0.16		0.42	0.17	
60+	434	0.37	0.28		0.35	0.22		0.43	0.16		0.38	0.18	
**Education**				<0.001			0.001			0.034			<0.001
≤Middle school license	545	0.37	0.28		0.37	0.21		0.45	0.17		0.39	0.18	
High school or diploma	932	0.41	0.27		0.40	0.20		0.44	0.17		0.41	0.17	
University or more	389	0.44	0.27		0.42	0.19		0.47	0.16		0.44	0.16	
**Volunteer**				<0.001			<0.001			<0.001			<0.001
No	923	0.38	0.28		0.34	0.19		0.42	0.17		0.37	0.16	
Yes	945	0.43	0.27		0.42	0.20		0.47	0.17		0.43	0.17	
**Odours Perception**				<0.001			<0.001			<0.001			<0.001
No	536	0.35	0.27		0.34	0.19		0.42	0.17		0.37	0.16	
Yes	1332	0.43	0.27		0.42	0.20		0.47	0.17		0.43	0.17	
**Exposure Area**				<0.001			<0.001			<0.001			<0.001
Control Area *	423	0.36	0.30		0.35	0.19		0.41	0.17		0.36	0.18	
Main Area **	1445	0.42	0.27		0.41	0.20		0.46	0.17		0.43	0.17	
**Residence Municipality**				<0.001			<0.001			<0.001			<0.001
**Main Area**													
Abbadia San Salvatore	455	0.42	0.26		0.43	0.20		0.48	0.15		0.44	0.16	
Arcidosso	284	0.44	0.27		0.42	0.20		0.45	0.17		0.43	0.17	
Castel del Piano	167	0.40	0.27		0.41	0.19		0.48	0.17		0.43	0.17	
Castell’Azzara	100	0.27	0.22		0.30	0.19		0.40	0.16		0.32	0.15	
Piancastagnaio	439	0.44	0.27		0.41	0.20		0.46	0.17		0.43	0.18	
Santa Fiora	225	0.40	0.25		0.38	0.18		0.42	0.17		0.40	0.16	
**Control Area**													
Castiglione d’Orcia	65	0.20	0.29		0.30	0.18		0.40	0.19		0.30	0.17	
Cinigiano	50	0.30	0.27		0.29	0.21		0.40	0.14		0.33	0.18	
Radicofani	43	0.36	0.42		0.27	0.21		0.42	0.14		0.30	0.21	
Seggiano	30	0.48	0.31		0.39	0.22		0.43	0.17		0.43	0.20	
**Occupational exposure to dusts, chemicals, pesticides, gas or radiation**				0.003			0.052			0.387			0.005
No	1110	0.42	0.28		0.40	0.20		0.45	0.17		0.42	0.17	
Yes	753	0.38	0.27		0.38	0.20		0.45	0.17		0.40	0.17	

Legend—N: number of questionnaires; HPI: Hazard Perception Index; EHPI: Exposure Hazard Perception Index; HRPI: Health Risk Perception Index; RPI: Risk Perception Index; SD: Standard Deviation; *p*: *p*-value. Notes: * Castiglione d’Orcia, Cinigiano, Radicofani, Santa Fiora, Seggiano; ** Abbadia San Salvatore, Piancastagnaio, Arcidosso, Castel del Piano, Castell’Azzara.

**Table 6 ijerph-19-14145-t006:** Ranking in descending order of the risk perception indices by municipality of residence.

Ranking Position	HPI	EHPI	HRPI	RPI
1st	Seggiano (C)	Abbadia San Salvatore (M)	Abbadia San Salvatore (M)	Abbadia San Salvatore (M)
2nd	Arcidosso (M) Piancastagnaio (M)	Arcidosso (MA)	Castel del Piano (M)	Arcidosso (M) Castel del Piano (M) Piancastagnaio (M) Seggiano (CA)
3rd	Abbadia San Salvatore (MA)	Castel del Piano (M) Piancastagnaio (M)	Piancastagnaio (M)	Santa Fiora (M)
4th	Castel del Piano (M) Santa Fiora (M)	Seggiano (C)	Arcidosso (M)	Cingiano (C)
5th	Radicofani (C)	Santa Fiora (M)	Seggiano (C)	Castell’Azzara (M)
6th	Cinigiano (C)	Castell’Azzara (M) Castiglione d’Orcia (C)	Radicofani (C) Santa Fiora (M)	Castiglione d’Orcia (C) Radicofani (C)
7th	Castell’Azzara (M)	Cinigiano (C)	Castell’Azzara (M) Castiglione d’Orcia (C) Cingiano (C)	
8th	Castiglione d’Orcia (C)	Radicofani (C)		

Legend: HPI: Hazard Perception Index; EHPI: Exposure Hazard Perception Index; HRPI: Health Risk Perception Index; RPI: Risk Perception Index; M: Main area; C: Control area.

**Table 7 ijerph-19-14145-t007:** Association among perception indices and question 12.5 “How do you judge the environmental situation of the municipality where you live in?”.

Environmental Situation	N	HPI		EHPI		HRPI		RPI	
		Mean	SD	Mean	SD	Mean	SD	Mean	SD
Excellent	125 (6.71%)	0.19	0.22	0.24	0.19	0.36	0.16	0.26	0.14
Acceptable	1057 (56.77%)	0.35	0.26	0.35	0.18	0.41	0.16	0.37	0.16
Serious but reversible	602 (32.33%)	0.52	0.26	0.49	0.18	0.53	0.15	0.51	0.15
Serious and irreversible	78 (4.19%)	0.58	0.23	0.54	0.18	0.55	0.18	0.56	0.15
*p*-value		<0.001		<0.001		<0.001		<0.001	

Legend—N: number of questionnaires; HPI: Hazard Perception Index; EHPI: Exposure Hazard Perception Index; HRPI: Health Risk Perception Index; RPI: Risk Perception Index; SD: Standard Deviation; *p*-value: observed probability of accepting the hypothesis of non-correlation between perception indices or vice versa the probability of rejecting the hypothesis of correlation.

**Table 8 ijerph-19-14145-t008:** Answers to question 12.5 “How do you judge the environmental situation of the municipality where you live in?” according to sex, age group, being a voluntary participant or not, belonging to main/control area and residing in one of the most exposed municipalities of the main area.

Factors		Environmental Situation—n (%)					
		Excellent	Acceptable	Serious but Reversible	Serious and Irreversible	Total	*p*-Value
**Sex** **(a)**	**Men**	82 (10.59)	447 (57.75)	220 (28.42)	25 (3.23)	774 (100)	<0.001
	**Women**	43 (3.95)	610 (56.07)	382 (35.11)	53 (4.87)	1088 (100)	
	**Total**	125 (6.71)	1057 (56.77)	602 (32.33)	78 (4.19)	1862 (100)	
**Age classes (years)** **(b)**	**18–39**	27 (5.58)	276 (57.02)	168 (34.71)	13 (2.69)	484 (100)	0.077
	**40–50**	24 (5.01)	273 (56.99)	163 (34.03)	19 (3.97)	479 (100)	
	**51–59**	37 (7.92)	257 (55.03)	151 (32.33)	22 (4.71)	467 (100)	
	**60–77**	37 (8.56)	251 (58.10)	120 (27.78)	24 (5.56)	432 (100)	
**Volunteer** **(c)**	**No**	67 (7.30)	584 (63.62)	234 (25.49)	33 (3.59)	918 (100)	<0.001
	**Yes**	58 (6.14)	473 (50.11)	368 (38.98)	45 (4.77)	944 (100)	
**Main/Control Area** **(d)**	**Main**	79 (5.49)	770 (53.47)	529 (36.74)	62 (4.31)	1440 (100)	<0.001
	**Control**	46 (10.90)	287 (68.01)	73 (17.30)	16 (3.79)	422 (100)	
**Main area** **(e)**	**Abbadia San Salvatore**	17 (3.75)	228 (50.33)	184 (40.62)	24 (5.30)	453 (100)	<0.001
	**Arcidosso**	14 (4.93)	126 (44.37)	132 (46.48)	12 (4.23)	284 (100)	
	**Castel del Piano**	15 (9.04)	86 (51.81)	55 (33.13)	10 (6.02)	166 (100)	
	**Castell’Azzara**	4 (8.00)	40 (80.00)	5 (10.00)	1 (2.00)	50 (100)	
	**Piancastagnaio**	11 (2.52)	261 (59.73)	150 (34.32)	15 (3.43)	437 (100)	
	**Santa Fiora**	19 (8.50)	140 (62.50)	55 (24.50)	10 (4.50)	224 (100)	

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
