# Peer review of "Community Concern about the Health Effects of Pollutants: Risk Perception in an Italian Geothermal Area"

_ijerph, 2022, doi:10.3390/ijerph192114145_

Round 1

Reviewer 1 Report (New Reviewer)

This is an interesting piece of research which is relevant to current discussions and largely well executed. 

I think the discussion session including some elements of weaknesses of the method could be added to a little. 

The survey did not ascertain length of residency or property owner ship (?)both of which are known to have some impact on environmental risk perceptions. Also whether respondents have children - again known to impact perceptions of risk given concerns about impact on child health. So - for example I would expect those with higher education and who own their property (ie potential negative impact on property value) and have young (under 16 years of age) children to express even higher levels of concern. 

I would also add that a qualitative element to follow up the survey research (perhaps through local focus groups) would be a beneficial next stage to help understand WHY people responded as they did. This would assist with future communication work. The research is very focused on individual perceptions but people share concerns/information/responses very much at a societal/community level. So we know that risk perceptions can become amplified in the social context. Social acceptance is referred to in the conclusions but this dimension is not introduced earlier.

Author Response

Reviewer 1

Comments and Suggestions for Authors

This is an interesting piece of research which is relevant to current discussions and largely well executed.

We thank the reviewer very much for appreciating our work and for identifying some weaknesses that could be improved. We will respond point by point by addressing the issues highlighted.

I think the discussion session including some elements of weaknesses of the method could be added to a little.

The survey did not ascertain length of residency or property owner ship (?)both of which are known to have some impact on environmental risk perceptions. Also whether respondents have children - again known to impact perceptions of risk given concerns about impact on child health. So - for example I would expect those with higher education and who own their property (ie potential negative impact on property value) and have young (under 16 years of age) children to express even higher levels of concern.

We agree with the reviewer and added this issue in the limitations and future perspectives at the end of the discussion section, as follows:

In addition, in the present study we did not consider the length of time the respondents have lived in the given area and whether or not they own the property, two factors known to have some impact on the perception of environmental risk. Also, whether respondents had/had not children, another known characteristic that affects risk perception because of concerns about the impact on children's health, was also not included among the variables of interest. This study has an exploratory nature, and it will be important for future research to analyse these issues in more detail.

I would also add that a qualitative element to follow up the survey research (perhaps through local focus groups) would be a beneficial next stage to help understand WHY people responded as they did. This would assist with future communication work. The research is very focused on individual perceptions but people share concerns/information/responses very much at a societal/community level. So we know that risk perceptions can become amplified in the social context.

We agree with the reviewer and added this issue in the conclusion, as follows:

For the future, it would be interesting to explore more specifically qualitative aspects, through activities based on local focus groups; this next phase of the research would be very useful to even better understand why people have a certain and specific perception of environmental risk related to geothermal energy and to the living context that people are experiencing. The present research is very focused on individual perceptions, but people share concerns/information/responses at the societal/community level and we know that risk perception can be amplified in the social context.

Social acceptance is referred to in the conclusions but this dimension is not introduced earlier.

We thank the reviewer for this observation. The issue of social acceptance is introduced at the end of the third paragraph of the discussion. To give prominence and logical basis to this argument, we have also included it in the sixth paragraph of the introduction section as follows:

Anthropogenic hazards seem to be more acceptable than natural hazards and are very related to social acceptance, knowledge and awareness of risk [24]. Social acceptance is crucial to address shared solutions based on trust in public authorities [25]. Therefore, it is valuable to share the results of scientific investigations, understand population perceived risks and engage communities to achieve sustainable environmental and health protection measures [17,19,26,27].

Reviewer 2 Report (New Reviewer)

Reviewer

In this article, the authors present a practical case study on the concern of society about the effects of pollutants on health applied to a region of Italy. It is considered that the topic could be of interest to the magazine's readers, but the work lacks some important information about the research. For all these reasons, a review of the text would be necessary to improve/refine certain aspects of the wording before proceeding to its possible publication.

My specific comments are as follows:

1º) Lines 40-43: It would be useful for the readers to clarify or indicate some information as to why all the geothermal power plants in Italy are located in the region of Tuscany.

2º) In figure 1 I suggest including the geographic north and the UTM coordinates of the area under study.

3º) I suggest including in point 2 a graph that explains the methodology carried out in the investigation and complements the indicated information. These charts are often very helpful to readers and make the research easier to understand.

4º) In section 2.4. entitled statistical analysis, (lines 226-227), the authors make the following reference “The comparison of the averages of the different perception indices among the study 226 factors categories”. Are you referring to the variance comparison method called ANOVA? I suggest clarifying this information regarding the method used. If this is the method used, I suggest including in section 3 the result of checking the normality of the groups used, since it could condition the result obtained. I also suggest including the R2 of the statistical results and not just the P value.

5º) Line 238: List the formula as 1.

Author Response

Reviewer 2

Comments and Suggestions for Authors

In this article, the authors present a practical case study on the concern of society about the effects of pollutants on health applied to a region of Italy. It is considered that the topic could be of interest to the magazine's readers, but the work lacks some important information about the research. For all these reasons, a review of the text would be necessary to improve/refine certain aspects of the wording before proceeding to its possible publication.

My specific comments are as follows:

1º) Lines 40-43: It would be useful for the readers to clarify or indicate some information as to why all the geothermal power plants in Italy are located in the region of Tuscany.

We thank the reviewer for this suggestion. Lines 40-43 have been modified as follows:

In Italy, all geothermal power plants in operation are located in Tuscany Region, the region that most of all represents Italian geothermal energy. The exploitation of this kind of energy began in the first half of the nineteenth century but in reality, historical sources have revealed that the heat of the earth and natural springs have been exploited since ancient times, at least for the entire first millennium BC by the Etruscan populations. Furthermore, these geothermal resources have proved to be among the most productive in the world [3,4]. Starting with Larderello, which today houses the largest geothermal plant in Europe, over the decades the number of regional geothermal plants has grown to more than thirty. Most of the Tuscan geothermal energy comes from the heat that derives from the intrusion of a magmatic pluto under the volcanic complex of Mount Amiata. The areas with abundant geothermal resources in Italy are many others, for example the regions such as Veneto, Friuli-Venezia Giulia, Campania, Sicily and Emilia Romagna; in any case, everything external to Tuscany has almost no relevance as an impact in absolute terms on the Italian energy balance.

2º) In figure 1 I suggest including the geographic north and the UTM coordinates of the area under study.

We thank the reviewer for this suggestion. We replaced Figure 1 with a figure reporting the information required.

3º) I suggest including in point 2 a graph that explains the methodology carried out in the investigation and complements the indicated information. These charts are often very helpful to readers and make the research easier to understand.

We thank the reviewer for this suggestion. At the end of paragraph 2.2, we added a new figure (Figure 2) reporting the methodology of the InVETTA project and how/where our study fits within the InVETTA project.

4º) In section 2.4. entitled statistical analysis, (lines 226-227), the authors make the following reference “The comparison of the averages of the different perception indices among the study factors categories”. Are you referring to the variance comparison method called ANOVA? I suggest clarifying this information regarding the method used. If this is the method used, I suggest including in section 3 the result of checking the normality of the groups used, since it could condition the result obtained. I also suggest including the R2 of the statistical results and not just the P value.

We thank the reviewer for this observations. The normality of the perception variables was tested and since the distributions of these variables were not normal the averages were compared using the non-paramteric Kruskal-Wallis equality-of-populations rank test. This information has been added in the methods section. Since we are interested in the statistical significance of the averages differences we then decided to report the p-value.

5º) Line 238: List the formula as 1.

We thank the reviewer for this suggestion. We believe it will be up to the editorial office to take care of this while editing the final version of the manuscript.

Round 2

Reviewer 2 Report (New Reviewer)

The authors have appropriately corrected the issues raised in my previous review report.

This manuscript is a resubmission of an earlier submission. The following is a list of the peer review reports and author responses from that submission.

Round 1

Reviewer 1 Report

Thank you for the opportunity to review your report on risk perception in communities in the geothermal energy production of Tuscany.  I have the following comments and suggestions.

Overall, the paper is too long. I think you can resolve this issue by condensing the introduction and the discussion sections. 

Introduction

The introduction has a lot of good background information but I suggest you focus on the objectives of your study and remove extraneous discussion. 

Lines 56-76: condense this discussion and consider summarizing the range of air concentrations/emissions of H2S, arsenic, and mercury from the geothermal plants since you specifically call out these constituents.  You don't need to discuss the abatement systems. 

Lines 83-87: excess mortality among men (particularly for liver and stomach cancer) in the Amiata area as compared to what? The male population of Italy as a whole? You cite a 2019 study by Nuvalone "confirming the excesses of mortality and hospitalization for respiratory diseases, also revealed a mortality decrease from ischemic heart disease and cerebrovascular diseases in in relation to high exposures to H2S emissions".  In the first study cited, you do not reference respiratory disease mortality and morbidity.  The title of the Nuvalone (2019) study indicates chronic exposure to low-level H2S.  Please check the validity of your statement. Also, what were the H2S air concentrations reported in Nuvalone (2019)? What do you consider "high exposures to H2S"? 

Lines 105-112: Please clarify - does this paper use the risk perception responses from the InVETTA project? Or are you reporting on an entirely new survey?  

Materials and Methods

What criteria did you use to select the municipalities most affected by geothermal emissions ("exposed group") as differentiated from your control group municipalities? Radial distance from a geothermal plant? Ambient air concentrations of H2S, mercury, and arsenic?  Why did you "over-sample" Piancastagnaio?

Please clarify my previous inquiry about your questionnaire and the data presented in this paper re: the InVETTA project.

I defer comments on the structure of your questionnaire and your statistical methods.

Results

I strongly suggest you organize your results summary tables by study area and control area.  Table 2. Arrange the residence municipalities by study and controls.  Clearly distinguish between study and control subject responses in all of your results tables. The way you currently present your results makes it difficult for the reader to follow your Discussion and do not support Conclusions.

Discussion

Reorganizing your Results section should help you to develop a more coherent discussion. What were the most important questions you wanted to answer with this study? Did the responses to your questionnaire answer the questions? Were there clear differences between Risk Perception Indices for respondents in the study area compared to the control area? Lines 396-399.

Lines 400-423: This appears to be more background information but does not directly support the data you are presenting. I suggest you delete this paragraph unless you can tie it directly to your study.

Lines 452-454: You state that (to your knowledge) this is the first study of risk perception related to geothermal energy.  I suggest you reorganize your discussion to clearly compare your major findings to other studies you reference in the preceding paragraph.  Where are your results consistent with other studies? Did you discover any new dimensions of risk perception that may be unique to geothermal energy?

Conclusions

As written, the Conclusions section does not summarize your key findings, why they are important, and how this study could be useful. 

Author Response

Comments and Suggestions for Authors

Thank you for the opportunity to review your report on risk perception in communities in the geothermal energy production of Tuscany. I have the following comments and suggestions.

Overall, the paper is too long. I think you can resolve this issue by condensing the introduction and the discussion sections.

Introduction

The introduction has a lot of good background information but I suggest you focus on the objectives of your study and remove extraneous discussion.

We thank the reviewer for the suggestion. We provided shortening the introduction and discussion sections.

Lines 56-76: condense this discussion and consider summarizing the range of air concentrations/emissions of H2S, arsenic, and mercury from the geothermal plants since you specifically call out these constituents. You don't need to discuss the abatement systems.

We thank the reviewer for this suggestion. We condensed the paragraph indicated (lines 56-76). Specifically, this paragraph was added to link up with the previous one in which was made a reference to the year 2019 Annual Report and we wanted to report the results of the monitoring activities relating to that year. Therefore, we have written the paragraph considering what has been previously said. The paragraph has been changed, adding a short paragraph from the cited report, as follows: “Particularly, monitoring activities are conducted by the Geothermal Sector of the Regional Agency for Environmental Protection of Tuscany (ARPAT) and all monitoring data are reported through annual reports (http://www.arpat.toscana.it/documentazione/report/report-geotermia/report-sul-monitoraggio-nelle-aree-geotermiche). Specifically, in 2019 no overruns of the emission limit values for the authorized parameters (mercury, hydrogen sulphide (H2S) and sulfur dioxide) were detected. The efficiency of ammonia and H2S abatement systems was higher than the minimum limit value [6].

Lines 83-87: excess mortality among men (particularly for liver and stomach cancer) in the Amiata area as compared to what? The male population of Italy as a whole? You cite a 2019 study by Nuvolone "confirming the excesses of mortality and hospitalization for respiratory diseases, also revealed a mortality decrease from ischemic heart disease and cerebrovascular diseases in in relation to high exposures to H2S emissions". In the first study cited, you do not reference respiratory disease mortality and morbidity. The title of the Nuvolone (2019) study indicates chronic exposure to low-level H2S. Please check the validity of your statement. Also, what were the H2S air concentrations reported in Nuvolone (2019)? What do you consider "high exposures to H2S"?

We thank the reviewer for these observations. We answered point by point.

Lines 105-112: Please clarify - does this paper use the risk perception responses from the InVETTA project? Or are you reporting on an entirely new survey?

We thank the reviewer for the opportunity to better clarify this point. We modified the text adding the following statement: “Particularly, here we are going to report and discuss results based on the InVETTA questionnaire on risk perception.

Materials and Methods

What criteria did you use to select the municipalities most affected by geothermal emissions ("exposed group") as differentiated from your control group municipalities? Radial distance from a geothermal plant? Ambient air concentrations of H2S, mercury, and arsenic? Why did you "over-sample" Piancastagnaio?

We thank the reviewer for the opportunity to better clarify this point. Considering that this survey focuses on the analysis of section 12 (risk perception) of the questionnaire used in the InVETTA project, the whole methodology refers to the aforementioned project. In our opinion, reporting the population selection and exposure assessment criteria would make the reading heavy and also would not be relevant to the main topic, so we considered it sufficient to insert the reference to the InVETTA project report. Anyway, we added the information regarding the criteria used to distinguish between the main area and the control area. As for the municipality of Piancastagnaio, this was oversampled because among all the municipalities of Amiata it is historically the one most exposed to emissions from the oldest and most polluting power plants. Anyway, in our opinion the statement “with oversampling for Piancastagnaio” is not useful to the text and therefore we decided to delete it. Therefore, the text was then changed as follows: “The study sample of the InVETTA project consisted in subjects aged 18-77 residing in the municipalities in the Amiata area most affected by geothermal plant emissions (Abbadia San Salvatore, Piancastagnaio, Arcidosso, Santa Fiora, Castel del Piano and Castell’Azzara defined as “Main area”) and in a control group of municipalities, always belonging to the geothermal area (Seggiano, Radicofani, Cinigiano and Castiglione d’Orcia), defined as “Control area”. The distinction between “Main area” and “Control area” is based on the atmospheric H2S concentration values estimated with dispersion models created using internationally standardized software and validated by the monitoring stations present in the area [9]. All further information on materials and methods of the InVETTA project can be found in the dedicated report published in 2021 [16].

Please clarify my previous inquiry about your questionnaire and the data presented in this paper re: the InVETTA project.

We thank the reviewer for this comment. We specified this concept in the introduction.

I defer comments on the structure of your questionnaire and your statistical methods.

Results

I strongly suggest you organize your results summary tables by study area and control area. Table 2. Arrange the residence municipalities by study and controls. Clearly distinguish between study and control subject responses in all of your results tables. The way you currently present your results makes it difficult for the reader to follow your Discussion and do not support Conclusions.

We thank the reviewer for this comment. As we stated at the end of the Introduction “The objective of the present paper is to develop risk perception indicators specifically related to the geothermal context, to study their characteristics, and to verify the association of these indicators with variables included in the questionnaire referring to socio-demographic characteristics, evaluation of the environmental situation, self-reported personal exposure to pollutants and place of residence.” (lines 136-140). The objective is not to evaluate the indices differences by comparing the main area with the control area, but to evaluate the differences between the indices with respect to some variables of the questionnaire. In fact, results and discussion are written in order to analyse differences among the indices respect some questionnaire variables. Specifically, two of the questionnaire variables are the "Residence municipality" and “belonging to the most exposed or least exposed area”. In our opinion, there’s no need to further distinguish between study and control responses. As suggested by the reviewer we arrange the residence municipalities by main and control area in Tables 2 and 5.

In addition, Table 8 has been modified to make it easier to read.

We also modified the Results paragraph.

Discussion

Reorganizing your Results section should help you to develop a more coherent discussion. What were the most important questions you wanted to answer with this study? Did the responses to your questionnaire answer the questions? Were there clear differences between Risk Perception Indices for respondents in the study area compared to the control area? Lines 396-399.

We thank the reviewer for these comments and the possibility to better clarify these points. As we stated before, the aim is not to compare indices among study subjects and control subjects, so we believe there is no need to assess “differences between Risk Perception Indices for respondents in the study area compared to the control area” as suggested by the reviewer.

Regarding the reviewer question “What were the most important questions you wanted to answer with this study?”, these question are stated in the aforementioned aim of the study. Furthermore, the risk perception questionnaire was built specifically to study various types of perception that people have on certain issues, in this case the presence of geothermal plants in the area.

The lines the reviewer refers to are: “Volunteers had a higher perception, as well as those who noticed bad smells around their homes. The sample living in the municipality of Abbadia San Salvatore, Arcidosso, Castel del Piano and Piancastagnaio showed a higher perception related to pollutants exposure.”

These results are reported in Table 5. As far as volunteers and who notice bad odours are concerned, this statement is supported by the results in Table 5. Volunteers (Volunteers – YES) have all the perception indices (HPI, EHPI, HRPI and RPI) higher than non-volunteers; also, who noticed bad smells around their homes (Odours perception – YES) had all the perception indices higher respect who did not noticed bad smells (please, refer to the extract of Table 5, reported below).

Volunteer

<0.001

<0.001

<0.001

<0.001

NO

923

0.38

0.28

0.34

0.19

0.42

0.17

0.37

0.16

YES

945

0.43

0.27

0.42

0.20

0.47

0.17

0.43

0.17

Odours Perception

<0.001

<0.001

<0.001

<0.001

NO

536

0.35

0.27

0.34

0.19

0.42

0.17

0.37

0.16

YES

1,332

0.43

0.27

0.42

0.20

0.47

0.17

0.43

0.17

We realized that the sentence "The sample living in the municipality of Abbadia San Salvatore, Arcidosso, Castel del Piano and Piancastagnaio showed a higher perception related to pollutants exposure" is incorrect and has been replaced with the following sentence: "The sampled subjects living in the municipalities of Abbadia San Salvatore, Arcidosso, Piancastagnaio and Seggiano, all belonging to the Main area except the last one, showed a higher perception."

This is supported by table 5 by looking at the "Residence municipality" factor, please refer to the part highlighted in green.

Residence Municipality

<0.001

<0.001

<0.001

<0.001

Main Area

Abbadia San Salvatore

455

0.42

0.26

0.43

0.20

0.48

0.15

0.44

0.16

Arcidosso

284

0.44

0.27

0.42

0.20

0.45

0.17

0.43

0.17

Castel del Piano

167

0.40

0.27

0.41

0.19

0.48

0.17

0.43

0.17

Castell’Azzara

100

0.27

0.22

0.30

0.19

0.40

0.16

0.32

0.15

Piancastagnaio

439

0.44

0.27

0.41

0.20

0.46

0.17

0.43

0.18

Santa Fiora

225

0.40

0.25

0.38

0.18

0.42

0.17

0.40

0.16

Control Area

Castiglione d’Orcia

65

0.20

0.29

0.30

0.18

0.40

0.19

0.30

0.17

Cinigiano

50

0.30

0.27

0.29

0.21

0.40

0.14

0.33

0.18

Radicofani

43

0.36

0.42

0.27

0.21

0.42

0.14

0.30

0.21

Seggiano

30

0.48

0.31

0.39

0.22

0.43

0.17

0.43

0.20

Lines 400-423: This appears to be more background information but does not directly support the data you are presenting. I suggest you delete this paragraph unless you can tie it directly to your study.

We thank the reviewer for this suggestion. We completely re-organized the Discussion paragraphs.

Lines 452-454: You state that (to your knowledge) this is the first study of risk perception related to geothermal energy. I suggest you reorganize your discussion to clearly compare your major findings to other studies you reference in the preceding paragraph. Where are your results consistent with other studies? Did you discover any new dimensions of risk perception that may be unique to geothermal energy?

We thank the reviewer for this comment. As we previously said, we completely re-organized the Discussion paragraph.

Conclusions

As written, the Conclusions section does not summarize your key findings, why they are important, and how this study could be useful.

We thank the reviewer for this comment; we completely re-organized the Conclusions paragraph.

Furthermore, in order to add the aim of the study, we modified the Abstract as follows: “Geothermal fluids for electricity and heat production have long been exploited in the Mt. Amiata area (Tuscany, Italy). Public concern about the health impact of geothermal plants has been present from the outset. Several factors influence the way people perceive risk, therefore the objective of the present research is to develop indicators of risk perception and assess indices differences in relation to some questionnaire variables. A cross-sectional survey was conducted in the Amiata area on 2,029 subjects aged 18–77. From the questionnaire section about risk perception from environmental hazards four indicators were developed and analysed. 64% percent of the subjects considered the environmental situation to be acceptable or excellent, 32% serious but reversible and 4% serious and irreversible; as the values of the various perception indicators increased, an upward trend was observed in the averages. Risk perception resulted higher among women and young people, and was associated with higher education. Those who smelled bad odours in their surroundings reported higher risk perception. Furthermore, risk perception was higher in 4 municipalities. The results represent the basis for further investigations, to analyse the link among RP indicators, exposure parameters and health status.”

Reviewer 2 Report

The work has an interesting title. Unfortunately, the authors conducted para-scientific research in which they use instruments that have not been validated, neither the sensitivity of the method nor its specificity is known. At work, there is no correlation between the actual threats to health and life of people and their fears. It is unclear whether the fears described are the primary medicine of man against the unknown or they result from real threats. The conclusions do not result from work, but are a political manifesto.

Author Response

Comments and Suggestions for Authors

The work has an interesting title. Unfortunately, the authors conducted para-scientific research in which they use instruments that have not been validated, neither the sensitivity of the method nor its specificity is known. At work, there is no correlation between the actual threats to health and life of people and their fears. It is unclear whether the fears described are the primary medicine of man against the unknown or they result from real threats. The conclusions do not result from work, but are a political manifesto.

We are sorry the reviewer did not appreciate our work. The study of risk perception (as part of risk and emergency management and communication) is an increasing scientific research field, including sociology and psychology sciences, medicine and environmental disciplines. The first studies dealing with the study of risk perception date back to the 1970s, and represent the integral part of risk assessment in the One Health perspective, as health is considered by WHO not only the lack of illnesses but a good physical and mental wellbeing.

Currently, in the scientific literature we can find many studies that analyse risk perception using analytical methodologies, such as that of Signorino and Beck, which we have taken as an example to calculate perception indices (quoted in the text; https://hal.archives-ouvertes.fr/hal-01092806/document). Furthermore, questionnaires are now widely used to study and analyse risk perception and the one used in the InVETTA study was obtained by reworking the questionnaires already used in other previously published scientific papers (quoted in the paper) and therefore validated by the scientific community. The journal we are proposing to publish the article dedicated a Special Issue to the Theme, "Research about Risk Perception in the Environmental Health Domain", and now it is open a second Special Issue.

As for the conclusions, we do not believe they can be defined as a political manifesto, but are instead a reflection from the present results and their relationship to previous literature in the specific area of environmental risk perception, as it is usually done in scientific papers. The conclusions complete the elements provided in the discussion. The text was changed to synthetize and clarify, as you will see.

We have made several changes to the manuscript hoping that it can be better understood by the readers.

In particular, we modified the text and the tables as follow:

  • we condensed the introduction and the discussion sections deleting some paragraphs;
  • we focused, in the new version, on the objectives of our study removing extraneous discussion;
  • we clarify the issue of the excess mortality in the Amiata area citing the 2019 study by Nuvolone and clarifying the concept of "high exposures to H2S";
  • we explained the criteria to select the municipalities most affected by geothermal emissions ("exposed group") as differentiated from the control group municipalities;
  • we reorganized our discussion and conclusions sections;
  • in the conclusions we summarized our key findings, why they are important, and how this study could be useful.

Furthermore, in order to add the aims of the study, we modified the Abstract as follows: “Geothermal fluids for electricity and heat production have long been exploited in the Mt. Amiata area (Tuscany, Italy). Public concern about the health impact of geothermal plants has been present from the outset. Several factors influence the way people perceive risk, therefore the objective of the present research is to develop indicators of risk perception and assess indices differences in relation to some questionnaire variables. A cross-sectional survey was conducted in the Amiata area on 2,029 subjects aged 18–77. From the questionnaire section about risk perception from environmental hazards four indicators were developed and analysed. 64% percent of the subjects considered the environmental situation to be acceptable or excellent, 32% serious but reversible and 4% serious and irreversible; as the values of the various perception indicators increased, an upward trend was observed in the averages. Risk perception resulted higher among women and young people, and was associated with higher education. Those who smelled bad odours in their surroundings reported higher risk perception. Furthermore, risk perception was higher in 4 municipalities. The results represent the basis for further investigations, to analyse the link among RP indicators, exposure parameters and health status.

We also would thank the reviewer for stimulating us to do a better paper and better expose our goals and results.

Round 2

Reviewer 1 Report

Thank you for your detailed response to my comments and questions, and for revising your manuscript accordingly.  I am fine with moving ahead with the revised manuscript.

Reviewer 2 Report

The work has an interesting title. Unfortunately, the authors research which they use instruments that have not been validated, neither the sensitivity of the method nor its specificity is known. At work, there is no correlation between the actual threats to health and life of people and their fears. It is unclear whether the fears described are the primary medicine of man against the unknown or they result from real threats.

Work is unstable and contains too many loops

In conclusion, if they are copyright, they should be published or they are published and you should not be in this part of the work only in the discussion.